# Examining the foreign policy attitudes in Moldova

Monica Răileanu Szeles *

Institute for Economic Forecasting, Transilvania University of Brasov, Brasov, Romania

* monica.szeles@unitbv.ro

## Abstract

This paper aims to examine the correlates of foreign policy attitudes in Moldova by a multi-level analysis, and to also reveal some characteristics of the Moldova's difficult geopolitical and economic context, such as the ethnical conflicts and poverty. A set of four foreign policy attitudes are explained upon individual- and regional level socio-economic and demographic correlates, of which poverty is the main focus, being represented here by several objective, subjective, uni- and multidimensional indicators. An indicator of deprivation is derived from a group of poverty indicators by the method Item Response Theory. Deprivation, subjective poverty, ethnicity and the Russian media influence are found to be associated with negative attitudes toward all foreign policies, while satisfaction with economic conditions in the country and a positive attitude toward refugees are both associated with positive attitudes toward all foreign policies.

## Introduction

The Republic of Moldova is situated in South-Eastern Europe, being at the confluence of Central Europe, Commonwealth of Independent States (CIS) and Balkans. The geographical position and the ethnical conflict embracing multiple forms has posed serious threats for this small multi-nation country, which had to face complex geopolitical challenges after the failure of the communist regime. In addition, the "multi-vector" external policy, permanently oscillating between East (European Union) and West (CIS countries), often characterized as "ambiguous, inconsistent and dual" [1], was not able to manage over time the ethnical and political conflicts.

The collapse of the former Soviet Union, Moldova's independence, the language law recognizing Romanian as the official language since 2013, the conflicts in Transnistria and Gagauzia, the perspective of unification with Romania, the Russia's influence and more recently the alternative of European Union membership are just few of the major political milestones that Moldova has encountered in the last 30 years.

The Moldova's perspective to join the EU could open a new chapter into its long term economic development. But the paradox is that Moldova is the only European country where European integration has progressively become less popular despite the pro-European government [2].

**Data Availability Statement:** All relevant data are within the manuscript and its Supporting information files.

**Funding:** The study is funded by the Erasmus+ Jean Monnet Chair programme. Jean Monnet Chair: Monica Raileanu Szeles Grant nr. 609500-

EPP-1-2019-1-RO-EPPJMO-CHAIR. Funder:
European Commission https://eacea.ec.europa.eu/
erasmus-plus/actions/jean-monnet_en. The
funders had no role in study design, data collection
and analysis, decision to publish, or preparation of
the manuscript.

**Competing interests:** The authors have declared
that no competing interests exist.

The closeness of Moldova to the European Union has historically been associated with the Moldova's fluctuant political regime. However, the struggle between the pro-European and pro-Russian parties has stacked for a long time Moldova between its neighbours, Romania and Ukraine. The European Union enlargement to the East, together with the emergence and development of Moldovan pro-European forces and opinions, should have been accelerated the EU membership. This hasn't happened, and moreover, the public support for the European integration has continuously declined after 2009. The failure of authorities to combat corruption, to increase the standard of living, and to prevent the exodus of the working age population explain the decline of the EU popularity because the population associates the pro-European government with the European integration process [2].

In Moldova, the ethnic identity has risen serious debates, as well as an overwhelming and ethno-political conflict, whose main actors are the Moldovans, Gagauzians and Romanians sharing one common country. According to the 2014 Census, the most important ethnic groups in Moldova are Romanians (7%), Ukrainians (6.6%), Gagauz (4.6%), Russians (4.1%) and Bulgarians (1.9%). After the Soviet Union disintegration, the largest group of Gagauzians form the Autonomous Territorial Unit (ATU) in Southern Moldova. Apart from other ethnic minorities in Moldova, Gagauz people have no other country bearing their name. The 1994 ATU autonomy law ensured that ATU will not become a part of Romania in case that Moldova will merge with Romania. This has quieted down Gagauzians for a while, but still ethnical and political tensions fuel fears of losing their autonomy.

The inconsistent foreign policy, the low public interest for current political issues, and the economic, social and political problems that Moldova encountered in the transition from communism to democracy [3], have downturned the long term economic development of Moldova, and have also deteriorated the strategic partnerships with neighbouring countries. To a much higher extent than other countries, the economic development of Moldova is significantly influenced by geopolitical forces and strategic partnerships, so that the foreign policy represents a fundamental pillar of Moldova's long term sustainable development.

Despite the rapid pro-poor economic growth of 5 percent annually since 2000, which resulted in a significant progress of poverty reduction from 68 to 11.4 percent between 2000 and 2014, Moldova remains one of the poorest countries in the region with 41% of population living below the threshold of 5 USD a day (2005 PPP) in 2014. I mention at this step that in the framework of economic theory the economic growth rate is considered to be pro-poor if it's the result of national policies aimed to use it for the benefit of poor people. This is the case here, as it resulted in the reduction of poverty. Moreover, the social disequilibria significantly inflate the negative impact of political instability and ethnical conflicts on economic development, which is strongly related to foreign policy in the case of Moldova. In this broad context, understanding what really lies behind the foreign policy attitudes, and in particular finding to what extent they are also influenced by poverty, ethnic tensions and regional patterns, allows enhancing the connection between public opinion and foreign policy, which could be ultimately regarded as an important assessment tool in measuring political commitment.

This paper intends to fill a gap in the literature by examining the foreign policy attitudes in Moldova, from a regional perspective. To capture not only the impact of individual-level characteristics on the foreign policy attitudes, but also the influence of geographical peculiarities, a multilevel model is used in the empirical section with individuals nested in districts. This multilevel approach reflects the regional perspective of our study.

The paper adds new empirical evidence to the literature which relies almost exclusively on studies of American foreign policy opinions [4], but it also contributes to the literature in other ways. First, it provides a regional perspective to the analysis of foreign policy attitudes, which perfectly fits the challenging Moldova's ethno-geopolitical context. Second, it relates

public opinions on foreign policy to poverty by a multidimensional approach, as to also address the social issues in Moldova—one of the poorest countries in Europe. Upon our knowledge, the link between foreign policy attitudes and poverty has not been explored so far. To accommodate the regional dimension of the dataset, random intercept logit models are used in the empirical section, where the regional foreign policy attitudes are explained upon regional- and individual level characteristics. In addition, the Item Response Theory is used to construct a scale of deprivation.

The paper is structured as follows: After "Introduction", the "Literature Review" underlines the most important contributions to the literature. The section of "Methodology" presents the two methods used in the empirical analysis, while "Data" provides the description of the data and the economic and social context in Moldova. The "Empirical analysis" includes the construction of the multidimensional scale of deprivation, and the analysis of foreign policy attitudes. The last section concludes and formulates policy recommendations.

## Literature review

The earliest strand of approaches to foreign policy attitudes emerged in the '50s as an echo of the American public opinion on both economic and military issues [5]. The Almond's seminal paper (1950) stating that "the foreign policy attitudes among most Americans lack intellectual structure and factual content" has been subsequently explored by most papers studying the foreign policy attitudes. This strand of literature commonly place all sets of attitudes on an internationalist-isolationist continuum [6–8]. In the context of the Vietnam War [9], identified for the first time internationalism as being the main vector of the American public opinion toward foreign policy. He explained that two faces of internationalism prevailed that time: the militant internationalism and the cooperative internationalism. The same approach was also explored by other papers, i.e. [8].

Cooperative internationalism focuses on achieving common goals through collaborative and non-military actions, as well as on being concerned about other countries and international issues [10]. In contrast with the Chittick's view [10, 11] analyzes cooperative internationalism as emerging from global solidarity and from the "obligation to the broader international community". Compared to cooperative internationalism, militant internationalism uses military strength and force in achieving foreign policy objectives [12], or simply as a consequence of self-defeating strategies. Isolationism is described as not overlapping with cooperative and militant internationalism, and it was often associated to nationalist unilateralism, e.g. [13].

As suggested above, the literature on foreign policy attitudes focuses almost exclusively on the US case study, and most conclusions and considerations are derived from the specific peculiarities of the US political context. For instance [14], explains that in general the foreign policy attitudes result from the perceptions of threat in relation with the Soviet Union military actions and intentions, given that the Soviet Union is generally perceived as a nation willing to expand its influence. An important strand of literature historically analyses the Soviet Union's image and perception in US compared to Europe, and most findings confirm that the US hostility is higher than that of European countries, even though the latter have stronger objective reasons to motivate it [15, 16]. Militant internationalism is therefore widely associated to the necessity to protect the American interests from the USSR in the past, and more recently from Islamic [17]. In the light of these aspects, when studying the foreign policy attitudes of a small country like Moldova that once was part of the Soviet Union, the literature mainly focusing on the US perspective could be irrelevant, and it might not fit the citizen profiles, thoughts and experiences. Despite the in-depth analysis of Americans' foreign policy attitudes, the literature

lacks the empirical research support of attitudes expressed by other citizens toward their national foreign policy.

Earlier and recent studies exploring foreign policy attitudes are all concerned with finding whether they are structured or not, and in case they are structured, what are their most important drivers, either political or core-values, or other kind of underlying factors. In the literature, the empirical evidence indicate mixed results. For instance [12], advance the idea that policy attitudes are structured upon core political values. More recent papers discuss the role of core political values in explaining the difference between policy preferences and opinions on the internationalism—isolationist dimensions, such as [18] and [19]. Given that they have "hidden organizing principles" [20], in empirical research values are not directly observable, so that they are measured as latent values by the factor loadings of the factor analysis. The foreign policy attitudes are also found to be connected to the "moral traditionalism" [21] in the sense that people who can be defined as being conservative on the moral traditionalism scale are more prone to be "militant anti-communist", and therefore to have an anti-Soviet sentiment.

Apart from the values driven attitudes discussed above, the analysis of the determinants of foreign policy attitudes is considered to be a difficult exercise because they are generally not based on information, are unstructured [22] and unstable [5], being therefore irrelevant for policy-making. In addition [23], argues that the quality, structure and coherence of public opinion on foreign policy depend on the level of knowledge. When analyzing the attitudes to issues in the area of European common foreign and security policy, he finds that their structure is low even at a relatively high level of knowledge.

As [12] emphasize, the foreign policy attitudes rely on postures or abstract believes about the international policy line that governments should follow, and to a lesser extent on the fully understanding of world politics. However [24], find that material and political considerations drive the Muslim citizens' attitudes toward the EU, and [25] explain the necessity of combining objective and subjective measures when studying the attitudes toward the EU.

When analysing the structure and drivers of votes and attitudes, the geographical polarisation should be also considered, and the traditional approach is the centre-periphery theory. The modern interpretation is provided by [26] who explain the difference between the citizens located in areas which are more connected to the global world, and those who are not, and the divide between the agglomerated urban centres with emerging knowledge economies, and the suburban economies or post-industrialized communities.

In the framework of the centre-periphery theory, the foreign policy attitudes have been also studied in relation with social position, and the seminal work belongs to [27], who developed the centre-periphery index—a summative measure of eight dichotomous items, such as sex, age, education, income, religion, sector of employment, occupational status, and urban-rural location. When analysing the foreign policy opinions in Norway upon the differences between the centre and the periphery of the society, he finds that the "periphery" will "either favour the status-quo, or sudden and complete changes". His influential idea was used to also prove that opposition to the EU increases linearly when moving from the centre to the periphery. In contrast with Galtung [28], proves the difference in attitudes between the centre and the periphery of society, but he doesn't identify the pattern of the difference [29]. explains the formation of political systems in Europe by the centre-periphery tension between the capital region and periphery regions, tension that was enhanced by the nation building process. The centre-periphery tensions also explain the formation of regional identities.

Apart from the foreign policy attitudes which are of interest for all countries in the world, the attitudes on unification and federalization are particularly significant only for a small number of countries, of which Korea and Germany are the most popular case studies. One of the

most prominent group of theories providing a consistent theoretical background for explaining the attitudes on unification is usually referred to as "generational theories". According to them, political values are formed in early adulthood based on the most important historical experiences [30–32], but the generational factors become "silent" during turbulent times, as advanced by [33, 34].

As discussed in this section, most papers studying foreign policy attitudes rely on the US case, and to a lesser extent on the study of other countries. Some kind of attitudes toward EU have been largely studies in the context of different referendums that were held over time on the topic of the EU or EMU (European Monetary Union) membership, and the findings generally exhibit a high degree of heterogeneity across the EU countries [35]. For example, the membership to EU is perceived as being economically beneficial for British people in the context of the Brexit referendum [25, 36]. Explain that the England's electorate attitudes toward the EU depend on the objective and subjective experiences of diversity and immigration, and therefore they conclude that objective and subjective measures should be combined when analysing political motivation.

The foreign policy attitudes of Moldovans have not been addressed in the literature so far, upon our knowledge, but few papers characterize the Moldova's foreign policy as "soft balancing" between Russia and European Union. More specifically [37], state that Moldova switched to the European Union using a light balancing strategy, but only when the EU has created the framework for such an attitude [38]. examine the similarities of the EU and Russia policies developed by them in Moldova, and they find that both aim attracting the local elites to be therefore able to indirectly influence the internal policy areas.

## Methods

The multilevel (hierarchical) analysis represents the principal methodology used in our empirical study, but subsequently the Item Response Theory will be used as well, as an alternative to the simple sum-score approach. In the empirical section of our paper the multilevel analysis allows us explaining a set of foreign policy attitudes upon a set of common individual- and regional level explanatory variables, of which the group of poverty variables are our main focus. In addition, the Item Response Theory (IRT) is used to aggregate a number of deprivation variables into a deprivation index which becomes one of the poverty variables.

The multilevel models are specifically designed for hierarchical data because they take into account the clustering of data upon different categories (Levels). Even though our data have a multilevel design (individuals at Level 1 nested in Districts at Level 2), in the first step of our empirical analysis we have to check the appropriateness of using multilevel models by running several specific tests that examine the degree of clustering for each level of our analysis. If our data are found to exhibit a significant degree of clustering, then the use of multilevel models becomes totally justified [39].

The two-level design of our data requests first examining the Interclass Correlation (ICC), and second, if the correlation is found to be reasonably high and justifies the use of multilevel models, examining what kind of multilevel models better fit our data. Ignoring that observations could be correlated at some Levels could lead to biased standard errors and incorrect results. The ICC is the most popular diagnostic-test for deciding on the appropriateness of the multilevel model. The fact that certain individuals live in the same district could cause their attitudes to foreign policy to be similar one from another, and to differ from those living in other districts. The ICC therefore measures the proportion of the total variation that is accounted for by the clustering of data, i.e. by between-districts variation.

The two-level variance components model is the simplest possible two-level model, so that it should be the first applied in our analysis. It is also referred to as the "empty model" because it has no covariates. The model is estimated by the maximum likelihood method.

$$y_{ij} = \beta_0 + u_j + e_{ij} \tag{1}$$

Where, $y_{ij}$ is the attitude of individual $i$ living in district $j$, $\beta_0$ denotes the intercept (the average across all individuals and districts), $u_j$ is the district level residual and $e_{ij}$ is the individual residual.

The district random intercept effects are assumed to be normally distributed with zero mean and between-district variance $\sigma_u^2$, and the individual' residuals also are normally distributed with zero mean and within-district variance $\sigma_e^2$.

Likelihood ratio (LR) tests are generally used when moving from a simple model to a more complex one, being indicative for the "badness of fit". The test also suggests whether it is worthy introducing a set of determinants, or whether the random-intercept model should be used instead of the variance components model. In our case the test allows us checking whether the "district random intercept effects" would be "value added" effects.

Another model that is used in the empirical section is the two level random-intercept model. The presentation of the two-level random intercept model, random-intercept logit model and random-intercept cumulative logit model is based on Steele (2010). As the model specified in Eq (1), this model has a fixed and a random part.

$$y_{ij} = \beta_0 + \beta_1 x_{ij} + u_j + e_{ij} \tag{2}$$

Where, $x_{ij}$ represent an individual level explanatory variable whose regression coefficient is $\beta_{ij}$. The model could easily incorporate more explanatory variables. The sum $\beta_0 + \beta_1 x_{ij}$ is the fixed part that could be extended by adding new explanatory variables, while the sum $u_j + e_{ij}$ is the random part.

However, in our analysis the dependent variables are ordinal and binary variables, so our specific multilevel models take the form of random intercept cumulative logit models and random intercept logit model, respectively.

The random intercept cumulative logit models can be written as in Eq (3).

$$\log(\Pr(y_{ij} \leq k) \,/\, \Pr(y_{ij} > k)) = \text{logit}(\gamma_{kij}) = \alpha_k + \beta x_{ij} + u_j, \quad k = 1, \ldots C - 1 \tag{3}$$

In Eq (3), the group level residual $u_j$, the cumulative response probabilities and the response probabilities allow the intercepts to vary across groups (in our case, districts). The parameter $\sigma_k$ is the overall intercept and it represents the log-odds that a person with $x = 0$ and $u = 0$ has a response of value equal to $k$ or lower than $k$.

The random intercept logit model accommodates binary response variables.

$$\text{Log}\,(\pi_{ij}/(1 - \pi_{ij})) = \text{logit}\,(\gamma_{kij}) = \beta_0 + \beta_1 x_{ij} + u_j, \tag{4}$$

Where, $\beta_0$ is the overall intercept and it represents the log-odds that $y_{ij} = 1$, when $x_{ij} = 0$ and $u_j = 0$. Consequently, the sum $\beta_0 + u_j$ is the intercept for the group $j$. The parameter $u_j$ is the random effect or Level2 residual. $\beta_1$ is the effect of the explanatory variable $x$ when holding constant the group effect, and it is also known as the cluster-specific effect.

The second method that we use here is the Item Response Theory, a measurement method traditionally and extensively used in psychometrics and educational sciences in the '60 and '70, but which has recently gained increased attention and was extended to all social sciences fields, as a group of mixed-effects multivariate generalized linear models. Compared to the Classical Test Theory, which simply calculate the mean of the item response scores, the IRT scores

describe the relationship between (1) the probability to give a certain response to an item and (2) the latent trait (summative score) and items characteristics (e.g. the difficulty and discrimination parameters), through a link function.

The IRT method is used here to assess the validity of the deprivation measurement scale that will be developed from a number of deprivation items available in our working dataset. The choice of this technique comes from considering deprivation as a latent trait, which is in line with the IRT method. Even though factorial analysis and test reliability have been advanced as alternative methods in the construction of latent traits based on a set of items, the IRT presents a number of advantages. For instance, with the IRT the conclusions are unveiled not solely based on the total scores, but rather on considering each item individually [40]. Other advantages are given by the IRT principles, such as the invariance of item parameters and individual's parameters, which allows comparing latent traits of persons of different populations. Another characteristic of this method is that it places both individuals and items on the same metric [41]. This helps selecting in the empirical section a group of deprivation items that are able to differentiate between individuals differently located on the deprivation scale.

In social sciences the one and two parameter- logit/ probit IRT models are the most popular IRT models [42]. We apply here the one parameter probit IRT model, which allows us analyzing the scale items upon their difficulty parameter, as well as calculating the individual deprivation scores. According to the notation used in mixed-effects regression models, the model can be written as:

$$V_{ij}^* = \beta_i + D_j^* + \varepsilon_{ij} \tag{5}$$

$$V_{ij} = 1 \ if \ V_{ij}^* > 0 \ and \ V_{ij} = 0 \ otherwise \tag{6}$$

In the equation above, $\beta i$ denotes the difficulty parameter for the item $i$, $Dj^*$ is the latent score of deprivation for individual $j$ and $Vij^*$ represents the response of individual $j$ to the item $i$. $Vij$ is a normally distributed error term with mean zero and fixed variance. If we treat $Dj^*$ as random individual effects, then the standard maximum likelihood provides estimates of both the parameter $\beta_i$ and the deprivation score $Dj^*$.

## Ethno-geopolitical context and data

### The substance of foreign policy attitudes in Moldova

The geopolitical deadlock between Russia and the EU, the unresolved conflicts over Transnistria, and the ethnic divide are the main coordinates on the Moldova's foreign policy agenda. The downward spiral of these drawbacks has unsystematically channelled an increasing wave of discontent toward the main actors of the Moldova's foreign policy, who are differently perceived by population either as catalysts or as opponents of the long term well-being and economic development.

In Moldova, the ethnic identity slides between Moldovanism and Romanianism which are different with regard to values, believes, political agendas and political goals [43]. Moldovanism advances the idea that Moldovans are different from Romanians, Romania being considered as a threat to Moldova's independence. Romanianism regards Moldova as a regional variation of the Romanian history and culture, the complementarity being in their view the link between Romania and Moldova. The debates emerging at every level of society around the Moldova's national identity have also fed the political conflict, with deep implications for the foreign policy too.

The foreign policy attitudes in Moldova should be therefore analysed in the context of the ethnical conflict and geopolitical context, which could in turn distort the population real attitudes prevailing in the absence of the above-mentioned tensions. The foreign policies examined in the empirical section will be shortly described below, just to provide a better understanding of their meaning, in the context underlined above.

Transnistria is a part of the Republic of Moldova officially recognized by Moldova as the Transnistria autonomous territorial unit with special legal status. In fact, Transnistria is a post-soviet frown conflict area, having an unresolved territory's political status in the sense that it is an unrecognised but de facto independent semi-presidential republic. The Transnistria war (1990–1992) emerged after the dissolution of Soviet Union between pro-Transnistria and pro-Moldova forces. Even though a ceasefire was declared in 1992, ethnic and political tensions have gradually accumulated over time. At present, a three-party (Russia, Moldova, Transnistria) Joint Control Commission supervises the security in the demilitarised zone, which is formed of twenty localities on both sides of the river.

Federalisation represents for Moldova more than an alternative form of local government. In the context of ethnic conflicts spread all over the country, federalisation can be rather considered a pro-Russian foreign policy measure that would weaken the country and would threaten the prospects of EU membership. This is so because federalisation would give Transnistria a bigger political power, which may tip the political balance to Moscow. In the framework of federalisation, the strengthening of district competences would facilitate international powers to influence and control the Moldovan districts.

In the light of the above considerations, the federalisation policy measure would be regarded as being in opposition with both the EU membership and the unification with Romania.

The NATO membership is a controversial project that is differently perceived in Moldova. Alienating with NATO, with or without joining it, could be a step toward the EU membership, but in the same time the NATO membership could violate the country's neutral status stated by Constitution. Due to the trustworthy information and the low quality of local media in Moldova, the opportunity and the "meaning" of NATO membership are insufficiently understood by population. As officially stated, NATO supports the Moldova's EU membership, and the NATO membership is not a precondition for the EU membership. From this consideration, the two policy measures are not in opposition one with another.

## Data

The empirical analysis uses data drawn from the 2017 wave of the Barometer of Public opinion in Moldova. This public opinion poll is a research program developed by the Moldovan Institute for Public Policy, on an annual basis, since 1998. The data collected by the Barometer cover areas like political choices, quality of life, and perception of economic, social and political measures adopted by the Moldovan government. The poll also represents an in-depth exploration of the attitudes toward (1) economic issues, (2) quality of life, (3) politics, (4) external relations, and (5) daily issues. The survey collects data on a number of 1103 adult individuals from all Moldovan districts, which represent the first-tier administrative- territorial units of Moldova. Although the first-tier administrative territorial units of Moldova include a number of 32 districts, only 31 are included in our analysis. Dubasari district is excluded, given that it is partially controlled by Transnistria, which is not considered in the current analysis. In addition, two municipalities (Chisinau and Balti), as well as the Autonomous Territorial Unit Gagauzia, are included in the working dataset.

The main focus of this paper is to explain the Moldovans *regional* foreign policy attitudes, so that four variables represent our main variables of interest, of which three are suggestive for the foreign policy, and another one for the domestic policy, the latter being taken as a reference model. The three foreign policies have been selected according to the literature, as to be representative for "militant internationalism", "cooperative internationalism", and "isolationism":

1. Attitudes toward EU membership (abbr. "EU membership"), as an indicator of the cooperative internationalism

2. Attitudes toward unification with Romania (abbr. "Unification with Romania"), as a "domestic" policy indicator

3. Attitudes toward federalization (abbr. "Federalization"), as an indicator of the isolationism

4. Attitudes toward the NATO membership (abbr. "NATO membership"), as an indicator of the militant internationalism

According to the Poll, 17% of the voting-age population would vote for federalization, 46% for the EU membership, 21% for the NATO membership, and 23% for the unification with Romania. A proportion of 25% are completely against federalization, 36% against the EU membership, 55% against the NATO membership, and 58% against the unification with Romania. It therefore results that the question about federalization has the highest number of undecided voters. The federalization question also exhibits the highest degree of dispersion among individuals, i.e. a value of 0.91, compared to 0.45 (Unification) and 0.50 (EU membership). Federalization is a categorical variable with three categories (pro-Federalization, against Federalization, and undecided), while the other two variables on the foreign policy attitudes are binary variables.

Our main variables of interest described above are explained in our paper upon a set of explanatory variables which includes both district- and individual- level variables, as follows:

1. Individual level variables

• Socio-demographic: age, age square, Moldovan, Gagauz, Working female, Education, Experience of living abroad (abbreviated "Abroad")

• Poverty: Income poverty, Deprivation, and Subjective poverty

• Information, attitudes and perceptions: Satisfaction with the current economic situation of Moldova (abbreviated "Satisfaction economy"), Satisfaction with Moldova foreign policy (abbreviated "Satisfaction foreign policy"), Communication with family, peers, friends and neighbors as the most trusty information source (abbreviated as "Social network"), Trust in Russian communication channels (abbreviated "Russian channels"), Information sources, Attitudes toward refugees, and Social trust

2. District level variables

• Rural-urban area of residence (dummy variables abbreviated "Urban")

• Region (three dummies—North, Centre and South)

Some of the explanatory variables were chosen according to the literature, while others have not been connected to the foreign policy attitudes so far, the latter reflecting thus the innovative contribution of this paper. The decision to introduce district level variables into our empirical analysis relies on the work of [26, 29], who find that geographical polarisation and the divide between urban and rural localities are drivers of attitudes and votes. Most of the individual-level correlates we consider here (age, gender-related variables, education, income,

occupation, urban-rural location, social position) have been advanced by [27], in his seminal paper, as items of the center-periphery index explaining the differences in attitudes toward foreign policy attitudes. In line with [25], we include subjective variables in the set of explanatory variables to reveal both objective and subjective experiences of diversity and immigration, which could be of a particular interest in a multi-ethnic country as Moldova. This justifies not only the use of subjective variables in the area of "Satisfaction" and "Poverty", but also of the variable "Experience of living abroad". As [24], we explain foreign policy attitudes upon material considerations (as reflected by the variable "Material deprivation"), and not only upon values and beliefs. The level of knowledge has been largely found to be a driver of attitudes [23], so that we examine here the attitudes toward foreign policy by education as well.

Most variables used to explain the Moldovans' attitudes toward different political issues have been recoded in variables of 2–4 categories. After recoding, "Education" is a categorical variable of three categories, "Income poverty" is a categorical variable comprising 12 income brackets, while Subjective poverty is a five category- variable giving insights into the level of satisfaction with the family income. Social trust is recoded as a dummy variable identifying the individuals who consider communication family, peers, friends and neighbors as being the most reliable source of information. The dummy variable "Satisfaction foreign policy" unveils the degree of satisfaction with the Moldova's foreign policy, while the dummy "Attitudes toward refugees" reflects the Moldovans' opinion about whether Moldova should provide or not protection to refugees. "Information sources" is a categorical variable of 7 categories: (1) Romanian TELEPHONE (considered as reference categories in the empirical section), (2) Russian TELEPHONE, (3) Moldovan newspapers, (4) Internet websites, (5) Social network (e.g. Facebook), (6) Announcements, posters, and (7) Communication with family, peers, friends and neighbors.

As explained in the empirical section, the latent variable "Deprivation" has been derived from a number of seven deprivation items, by using the Item Response Theory. The items are: Refrigerator (item 1), TELEPHONE (item 2), Computer (item 3), Car (item 4), Washing machine (item 5), Current water (item 6) and Gas (item 7). All items belong to the areas of durable goods and utilities. The means and standard deviations reported in Table 1 reflect a high heterogeneity across population with regard to the deprivation rates corresponding to the scale items.

Given that the regional dimension represents a primary focus here, the variables of interest will be represented by regional maps. They are plotted on Moldova's map, using district level data. Different shades of grey describe the Moldovan' attitudes toward the external relations of Moldova by districts. The white coloured districts indicate a higher concentration of those who are against that political measure, while a black area is suggestive for a high proportion of population who is in favour of that political issue.

**Table 1. Deprivation items: Summary statistics.**

| Deprivation item | Mean | Standard deviation |
|---|---|---|
| Refrigerator | 0.04 | 0.21 |
| Telephone | 0.03 | 0.18 |
| Computer | 0.38 | 0.48 |
| Car | 0.59 | 0.49 |
| Washing machine | 0.18 | 0.38 |
| Current water | 0.25 | 0.44 |
| Gas | 0.35 | 0.47 |

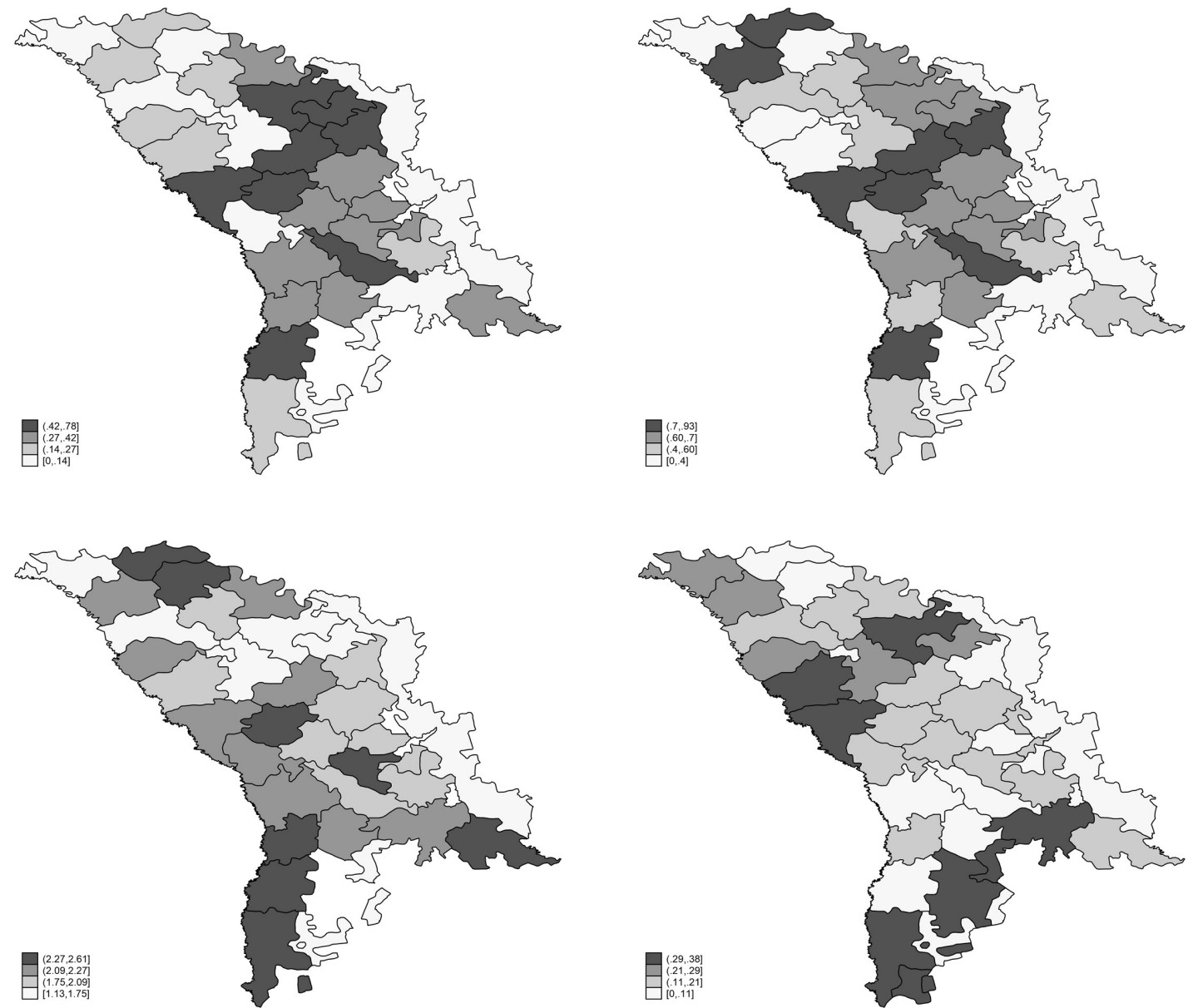

**Fig 1. 1.1 Unification with Romania; 1.2 EU membership; 1.3 Federalization; 1.4 NATO membership.** *Notes.* (1) The maps were realized by author in Stata using the GIS mapping; (2) Transnistria is excluded from our analysis.

As shown in Fig 1, the maps of the territorial distribution of votes exhibit some degree of similarity. In particular, the lower half of Fig 1(1.1) and 1(1.2) are very similar. The South-Eastern half of Moldova tends to be more reluctant to all foreign policy solutions (and to a lower extent to the domestic policy measure of federalization). However, the perspectives of federalization and NATO membership seem to raise more negative votes than the EU membership and Unification with Romania. Surprisingly, the Moldova's Western border with Romania is not characterized by a higher concentration of pro-unification and pro-EU membership votes. Only the South-Eastern part of Moldova appear to be "darker", which means a higher proportion of votes for the unification with Romania and EU membership. Moreover, the four maps indicate some regional patterns, and this will be analysed in the next section.

## Empirical analysis

The empirical analysis is conducted in two steps. First, a score of deprivation is calculated based on a group of deprivation variables by the IRT method. Second, the deprivation score, along with other poverty indicators and other covariates, will explain a set of foreign and domestic policy measures, in order to firstly reveal whether poverty, and subsequently what kind of poverty, influence the citizen' attitudes toward foreign policy measures. The IRT is therefore used in the first part of the empirical analysis, and two-level random-intercept cumulative logit models in the second part.

## Deriving a latent score of material deprivation by the Item Response Theory

The central objective of the paper is to examine the impact of poverty on the Moldovans' foreign-policy attitudes. As widely debated in the literature, poverty is a broad concept that has been largely addressed by various approaches, such as the uni- versus multidimensional, and the objective versus subjective ones. Given the comprehensive area of poverty, three indicators are used here to operationalize this broad concept. "Income poverty" and "Subjective poverty" are directly taken from the dataset, while the third, "Deprivation" will be calculated in this section. "Income poverty" measures the monetary unidimensional poverty, while "Deprivation" represents the multidimensional poverty. Both indicators reflect the objective nature of poverty, which is in contrast with "Subjective poverty". These three indicators allows us incorporating a multifaceted representation of poverty in our empirical analysis.

As a latent measure, "Deprivation" will be derived from a set of indicators reflecting the economic strain. The method used at this step is the Item Response Theory (IRT) because it justifies a rigorous selection of deprivation items, and it also allows calculating a latent score of deprivation.

The variables used to operationalize deprivation reflect two dimensions of material deprivation: Housing dimension and Durable goods dimension, as denoted in the Eurostat database (Eurostat, 2018).

- Durable goods dimension: Refrigerator (item 1), Telephone (item 2), Computer (item 3), Car (item 4), and Washing machine (item 5);

- Housing facilities dimension: Current water (item 6) and Gas (item 7).

We choose here the one-parameter probit IRT model to construct the scale of deprivation and to calculate the individual deprivation scores. The selection of items is done according to the item difficulty parameters, as the item discrimination parameters are fixed in this model. A broad range of item difficulty parameters would be an indication of a comprehensive scale. The item parameter invariance is the key-feature of the IRT and makes the distinction between the IRT and CTT. For our analysis this propriety means that the item estimates do not depend on person samples and persons estimates do not depend of item samples. It therefore ensure the consistency of the individual scores of deprivation.

In a first step, the Cronbach's Alpha test is applied as to check the internal reliability of the deprivation scale. Even though several sets of items have been tested and analysed using the Cronbach's Alpha and the IRT, only the final selected one is reported here. The value of 0.69 denotes an acceptable internal consistency and it also proves that the items included in the scale measure the same latent phenomenon—material deprivation.

In Table 2 the material deprivation items are ranked upon their difficulty parameters. According to our data, the refrigerator and telephone are the most difficult or severe items

**Table 2. Estimates from the one-logit IRT model.**

| Deprivation item | Difficulty parameter (coefficient/ Standard error) |
|---|---|
| Car | -0.56*** (0.09) |
| Computer | 0.73*** (0.09) |
| Gas | 0.90*** (0.09) |
| Running water | 1.57*** (0.10) |
| Washing machine | 2.21*** (0.11) |
| Refrigerator | 4.21*** (0.18) |
| Telephone | 4.63*** (0.21) |

Notes. Estimates from the one-parameter probit IRT;

*** p<0.01,

** p<0.05,

* p<0.1.

which suggests that a person who has no refrigerator or telephone has a high probability (higher than 0.5) to be also deprived of other items. In fact, the item difficulty parameter is the value along the "material deprivation" continuum at which an individual has 0.5 probability of being deprived.

The Item Characteristic Curve (ICC) in Fig 2 describes the probability of being deprived upon a certain deprivation item conditional on specific values of the deprivation distribution. The difficulty of a deprivation item, as a location index, describes where the item functions along the deprivation scale. Given that all items have the same discrimination power, their ICC do not intersect as it would be in the two-parameter model.

Under the IRT theory, the left-hand curves represent "easy" items, the centre-curves represent items of medium difficulty, while right-hand curves represent "difficult" items. For instance, "Refrigerator" and "Telephone" are "difficult" items because the probability to be deprived on these items is generally low even for the deprived individuals, while "car" is a "the least difficult" item because the probability of being deprived on this item is high for most deprived individuals.

### Examining the correlates of foreign policy attitudes and the impact of poverty. Analysis and discussion

The empirical analysis is intended to exploit the two-level structure of the data by using the multilevel analysis with individuals at Level 1 nested in districts at Level 2. This allows us using both individual- and district- level variables while also considering the correlation at each level of analysis.

In the first part of this section we check whether this technique is appropriate and fits our data by the variance components model. Although this "preliminary analysis" is separately performed on each of our four variables on foreign/domestic policy attitudes, the results from the variance components model are extensively reported and analyzed here just for the variable "EU membership". However, the conclusions of this part will regard the whole set of four variables.

The core of the preliminary analysis is the identification of the variation at each Level of analysis with the Variation Partition Coefficient (also called "Interclass Correlation Coefficient"). The Variance Partition Coefficient (VPC) is the proportion of the total residual variance (Level 1 + Level 2) that is due to between-group variation. The VPC for the variable "EU

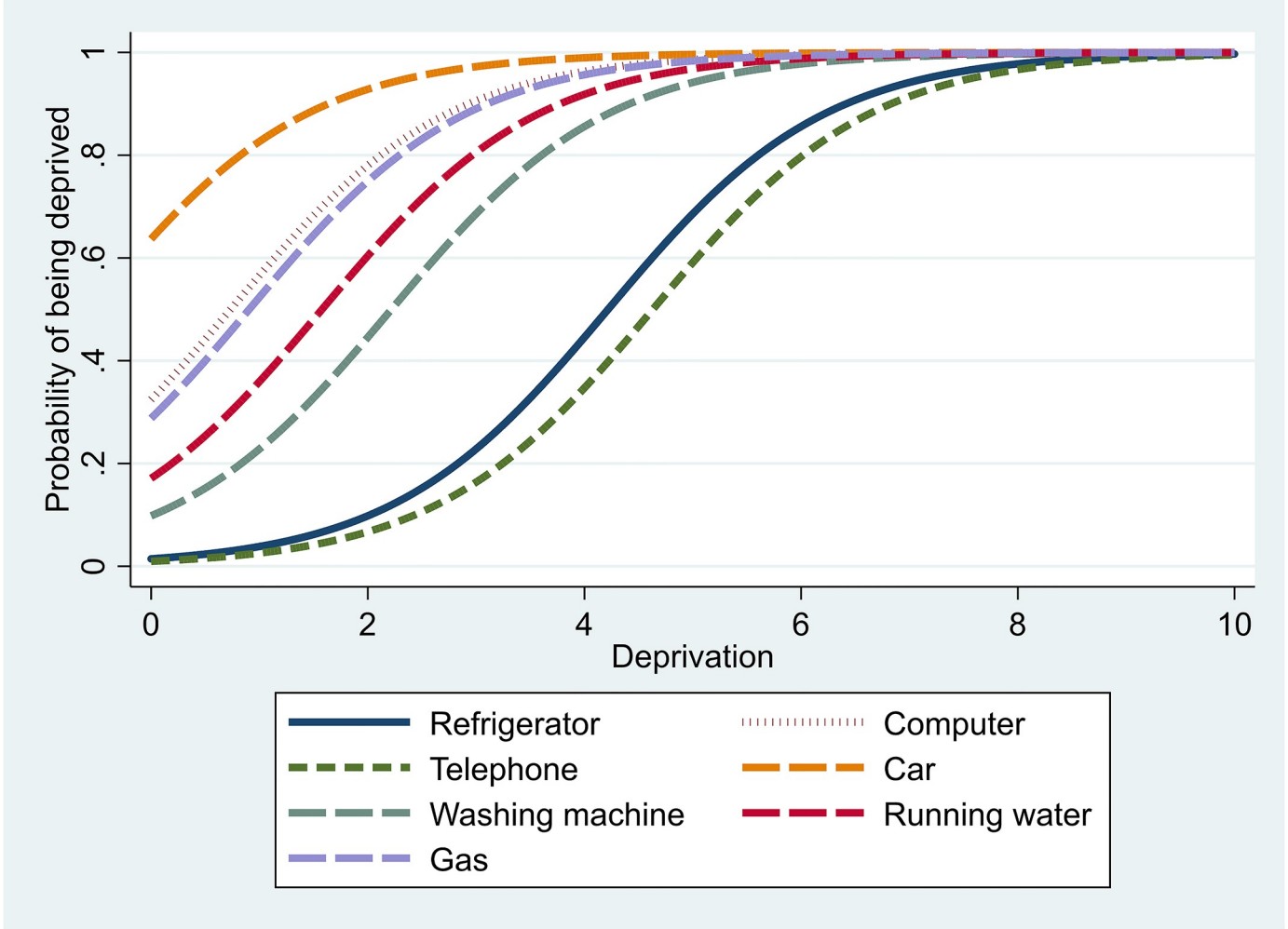

**Fig 2. Item characteristic curves—Material deprivation items.** The ICC graphical representation indicates a balanced selection of deprivation items in terms of their difficulty. Beside the item difficulty parameters, the IRT method allows calculating the latent scores of material deprivation. The individuals' scores of material deprivation enter as explanatory variables in the second part of the empirical analysis, but they also allow analyzing the regional distribution of deprivation scores, as shown in Fig 3.

membership" is analyzed by the two-level logit model with district random effects, but no explanatory variables. This is a 'null' model, which is also referred to in the literature as a variance components model. The output generated by this model can be synthetically presented as follows: The log-odds of voting Moldova's accession to EU for an 'average' locality (with $u_0$ = 0) is estimated as 0.30. The intercept for locality $j$ is $0.30+u_{0j}$, where the variance of $u_{0j}$ is estimated as 1.728. The likelihood ratio statistic for testing the null hypothesis is 85.46 with a corresponding p-value of less than 0.00005, and so there is strong evidence that the between-district variance is non-zero.

When plotting the estimated residuals for districts, we get that, for a substantial number of communities, the 95% confidence interval does not overlap the horizontal line at zero, indicating that the votes of Moldova's accession to EU in these districts is significantly above average (above the zero line) or below average (below the zero line). This also indicates the significance of district effects from the null model.

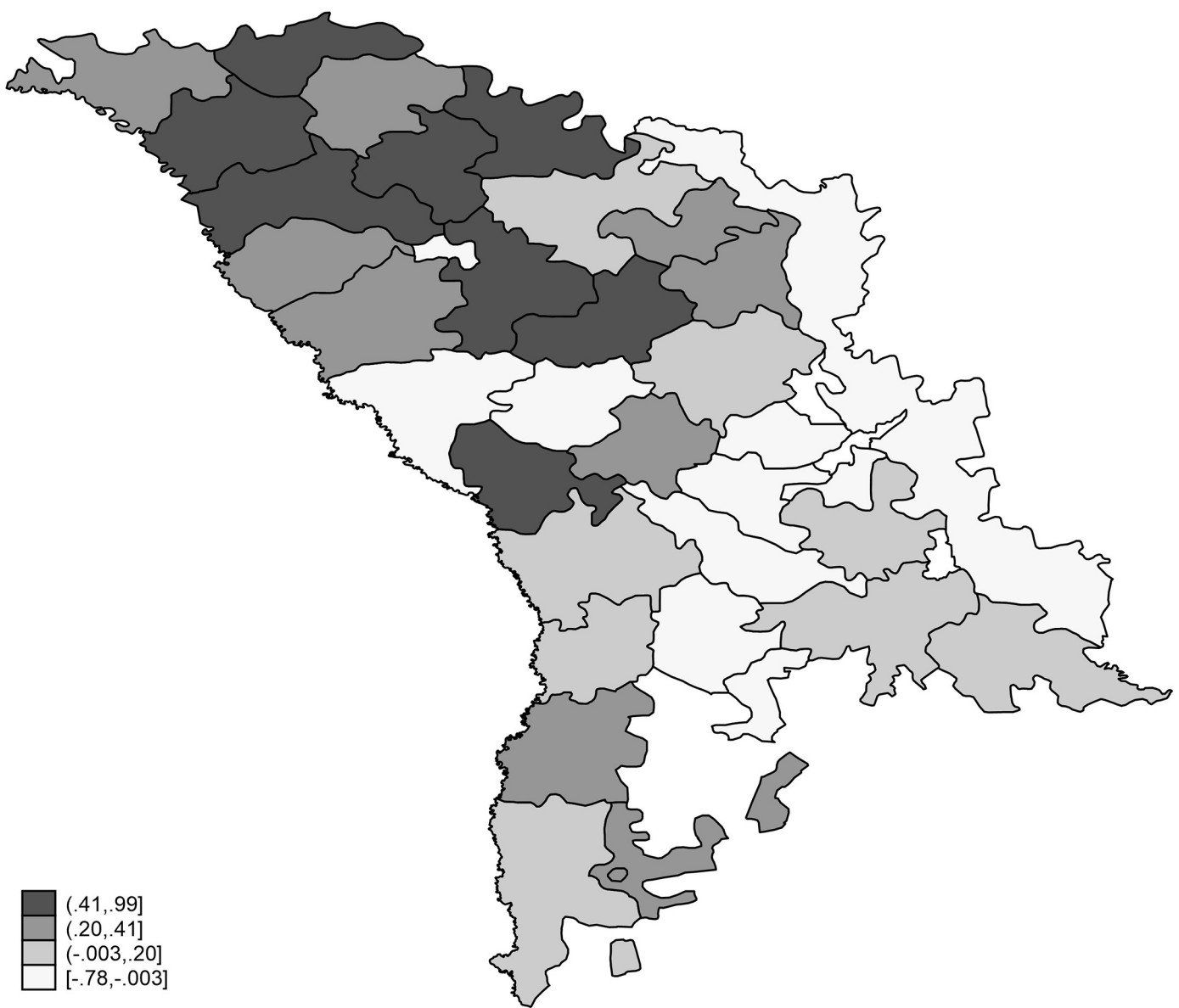

(.41,.99]
(.20,.41]
(-.003,.20]
[-.78,-.003]

**Fig 3. The regional distribution of material deprivation in Moldova.** Source. The map was realized by author in Stata using the GIS mapping. In Fig 3, the darker areas reflect the most deprived districts, while the lighter ones are associated to a lower material deprivation. Apparently, the Northern part of Moldova has more deprived districts compared to the Southern part. However, the regional heterogeneity seems to be an essential characteristic when examining the regional material deprivation in Moldova.

In the "null" variance components model presented above, which estimates the grand mean of the votes for the Moldova's accession to EU, the between-district variance is estimated as 1.72. For a *logit* model, the Level 1 residuals are assumed to follow a standard logistic distribution which has a variance of $\pi^2/3 \approx 3.29$. This implies a VPC of the district level for the Moldova's accession to EU votes equal to $1.72/ (1.72 + 3.29) = 0.3433$. This value allows us concluding that 34.33% of the variation in the Moldova's accession to EU votes is due to between-district variation. Similarly, we get that the VPC for the pro-unification with Romania votes is 30%, the VPC for the NATO membership votes is 16%, while the VPC in the pro-federalization votes is 15%.

To sum up the preliminary analysis undertaken in this section, we conclude that the size of variation at Level 2, which is higher than 10%, justify the use of mixed models instead of linear ones for all our variables of foreign/domestic policy attitudes.

In Table 3 the two-level random intercept (cumulative) logit model has been used to explain the four variables of foreign/domestic policy attitudes upon a common set of individual- and regional level explanatory variables, of which a group of three poverty variables represents our main focus.

According models (1), (2) and (4), younger Moldovan people seem to be more in favour of foreign policies involving international cooperation and openness, as those examined here. Although the results in the literature are rather mixed, our findings confirm the general consideration that intellectuals and the younger generation of leaders questioned the concept of a distinct Moldovan nation, and reemphasized the Romanian identity, in contrast to older generations that are rather Russian-born [44]. In general, age is associated in the literature with

**Table 3. Explaining foreign and domestic policy attitudes upon individual- and district level variables.**

| Explanatory variables | EU membership (Random intercept) | Unification with Romania (Random intercept) | Federalization (Random intercept) | NATO membership (Random intercept) |
|---|---|---|---|---|
| | Model 1 | Model 2 | Model 3 | Model 4 |
| **Individual level** | | | | |
| **Socio-demo** | | | | |
| Age | -0.13*** (0.03) | -0.13** (0.03) | -0.03 (0.02) | -0.10*** (0.03) |
| Age square | 0.001*** (0.0003) | 0.001** (0.0003) | 0.0003 (0.0002) | 0.001*** (0.0003) |
| Working female | 0.43** (0.24) | 0.43* (0.24) | 0.36** (0.18) | -0.05 (0.23) |
| Education | 0.31** (0.14) | 0.26* (0.16) | 0.32*** (0.11) | 0.17 (0.15) |
| Living abroad | -0.002 (0.20) | -0.22 (0.22) | -0.06 (0.15) | 0.001 (0.01) |
| Gagauz | -2.60*** (0.84) | -3.09*** (1.14) | -1.08*** (0.43) | -1.90*** (0.80) |
| Moldovan | 0.66*** (0.27) | 0.09 (0.29) | 0.25 (0.20) | 0.25 (0.27) |
| **Poverty** | | | | |
| Deprivation | -0.80*** (0.17) | -0.91*** (0.19) | -0.15 (0.12) | -0.80*** (0.17) |
| Subjective poverty | -0.23* (0.13) | -0.33*** (0.13) | 0.23** (0.10) | - |
| Income poverty | 0.29 (0.29) | 0.25 (0.31) | 0.11 (0.20) | 0.40 (0.28) |
| **Satisfaction** | | | | |
| Satisfaction economy | 0.61*** (0.14) | 0.37*** (0.15) | 0.49*** (0.10) | -0.02 (0.13) |
| Satisfaction foreign policy | 0.11 (0.13) | 0.26** (0.13) | -0.23*** (0.09) | 0.24** (0.12) |
| Trust in people | -0.44* (0.27) | -0.49* (0.30) | 0.20 (0.20) | |
| Attitudes toward refugees | 0.57*** (0.21) | 0.61*** (0.21) | 0.35** (0.15) | 0.38** (0.19) |
| Russian channels | -1.54*** (0.32) | -1.40*** (0.42) | -1.15*** (0.24) | -0.59* (0.36) |
| **Regional level** | | | | |
| Region | | | | |
| Centre | 0.31 (0.38) | 0.72** (0.34) | 0.32 (0.21) | - |
| South | -1.00** (0.45) | -0.14 (0.42) | 0.70*** (0.26) | |
| Rural | 0.63 (0.41) | 0.88** (0.38) | 0.16 (0.21) | 0.31 (0.21) |

Notes. Random intercept models; The estimation uses a mean-variance adaptive Gauss-Hermite quadrature with 7 integration points for each set of random effects; The estimation procedure is the maximum likelihood estimation using adaptive quadrature with 7 integration points; The standard errors are reported in brackets;

Reference category for Region is North;

*** p<0.01,

** p<0.05,

* p<0.1.

increasing political knowledge even when considering the differences in the educational attainments [45–47]. But according to the generation effect theory, in (former) transitional economies the political preferences are usually influenced by the periods of dissatisfaction with a certain political regime, so that younger people are more progressive than older ones in terms of values [48]. Our findings are therefore in line [48], in the sense that in Moldova (a former transitional country) younger people are more supportive of international cooperation in comparison with older generations. Interestingly, age is not significantly associated to federalization—the domestic policy measure considered here as a reference one, for comparative purpose. The significance of the quadratic term of age (age square) suggests a nonlinear relationship between age and the dependent variables in Models (1), (2), and (4). The positive coefficients in quadratic term indicates in fact that the relationship between age and the variables of interest is of a convex form.

In comparison with women outside the labour market, working women are significantly found to be more open to both foreign policies (EU membership and unification with Romania), as well as to the federalization policy measure, but against the NATO membership. This clearly suggest their positive attitude toward "cooperative internationalism" with EU and Romania (the Western neighbours), and the rejection of "militant internationalism" (reflected here by the NATO membership). Our finding is supported by previous papers, such as [49].

In comparison with lower levels of educational attainments, individuals with higher levels are found to be more supportive for the "cooperative internationalism" and federalization, but not for "militant internationalism" as well. The result is largely confirmed by the empirical evidence collected over time from US. Different channels have been advanced in the literature as mechanisms explaining the relationship between education and (foreign) policy attitudes or political trust.

Higher educated people have been found to generally support internationalist policies, while the less educated have been identified as supporting the isolationist policies [50]. Despite the rich empirical evidence suggesting that the highly educated were in favour of militant internationalism in the 1970s, this association has weakened over time. Still, some evidence confirms that education is directly associated to the militant internationalism, and indirectly related to cooperative internationalism. In contrast [51], shows that those having college educational attainments support the cooperative internationalism stronger than the militant internationalism. Other papers suggest that education experience help the development of political interest that is generally shaped by the class differences [52]. Education is also found to be the most important predictor of political knowledge, even after accounting for personality traits and intelligence [53].

The positive impact of education in Models (1)–(3) can be also explained by the strong correlation between education and political trust [54]. From a different perspective, our results are in line with the "centre-periphery theory" [27] according to which a low level of education indicates a "periphery" position which is associated to the opposition to the EU, as also indicated by our results.

Despite our expectation that people who have lived abroad for a while are more open to internationalization, or at least have stronger beliefs in the area of foreign policy, being influenced by the EU or Russia political spectrums, the EU and Russia being the most popular destinations for Moldovan immigrants, this hypothesis was not confirmed by the data analysis.

The Gagauz people are a Turkic people living mostly in a Sothern autonomous region of Moldova, officially called the Autonomous Territorial Unit of Gagauzia. In the framework of the 2014 referendum that took place in Gagauzia, 98.4% of voters chose to join the Eurasian Union, supported and promoted by Russia, instead of the EU. The regression coefficients indicate that Gagauz people, who represent a small ethnic minority in Moldova, have significant

and strong negative attitudes toward all four policy measures considered in our model. This is according to our expectations, and it relies on the regional political conflicts running for decades in the ATU, as well as on the fear that the Moldovan Gagauz will end up in "a new version of greater Romania" [44]. Nevertheless, the specific political context in which Gagauz people have lived over time under the influence of surrounding countries, justifies their opposition to any significant political change in Moldova that directly or indirectly could threaten their autonomy as well. The literature also confirms the opposition of "distinctive ethnic identities" to the militant internationalism [49], which supports our finding in Model 3.

According to our results, in spite of Moldova's oscillation between the Russia way and European Union path, Moldovans are found to be in favour of the Moldova's EU membership. However, the NATO membership is not a seen as an appropriate foreign policy measure for Moldova, not by Moldovans, and nor by Gagauz people, and one explanation could be the strong Russian opposition [55].

For a comprehensive conceptual delimitation, the influence of poverty is analysed by three variables which are indicative for both the objective and subjective poverty. Objective poverty is reflected by two indicators, i.e. income poverty and deprivation, and subjective poverty is reflected by a categorical variable. While income poverty is found to have no significant effect on our policy measures, deprivation is found to be a significant correlate of all foreign policy attitudes in the sense that the deprived people are more likely to be against all three foreign policies, even though the policies do not overlap each other. In turn, deprivation has no significant impact on federalization—the reference domestic policy measure. The relationship between deprivation and foreign policy attitudes has not been addressed so far in the literature upon our knowledge, but the links between material stratification and political consciousness have been assessed as fragile and variable [47].

People who perceive themselves as being income poor are against the EU membership and also against the unification with Romania, but they would agree on the Moldova's federalization. In fact our results suggest that, in contrast with the people objectively identified as income poor, those suffering from deprivation and subjective poverty are in opposition with all policy measures involving openness and internationalization. Even though the relationship between poverty and foreign policy attitudes has not been addressed so far in the literature, our empirical findings could rely on the "centre-periphery theory" [27] which states that periphery is "parochial" and "sceptical" on issues reaching beyond national concern.

Another category of explanatory variables are from the area of satisfaction. People satisfied with the Moldova's economic situation are more likely to agree with all policy measures, excepting the NATO membership. It is interesting to note that those who declare to be satisfied with the foreign policy in general, would vote for the Moldova's unification with Romania and NATO membership, but against federalization. The level of social trust, represented here by the variable "Trust in people", is found to be negatively correlated with the votes for the EU membership and unification with Romania. This is in contrast with the literature which considers social trust as being associated with social participation and engagement in the society, and therefore as a driver for democratic and efficient governments [56]. However, our finding should be interpreted in the context of the interpersonal trust in Moldova being among the lowest in Europe, i.e. 3.7 times lower compared to that of Sweden, and the lowest of the post-communist countries [57].

Despite the fact that Moldova is a country of mass emigration, to Russia and Ukraine for short-term periods and to the EU for long term staying, it is interesting to capture the Moldovan's attitudes toward refugees and to see whether it is related to the attitudes toward foreign policies. We expect to find that people who are in favour of the right for people to seek refuge should also support the Moldova's EU membership, as the EU is an effective actor for peace

and democracy. In a broader perspective we expect to get a strong relationship between a positive attitude toward refugees and o positive attitude to all types of internationalisation foreign policies. The empirical results confirm that our expectations are met. The attitudes toward refugees also reflect the Moldovans' opinion about whether Moldova should provide or not protection to refugees. Our results show that those who support the idea that Moldova should help refugees, also support all political measures, i.e. the both types of cooperative and militant internationalism policies. This empirical finding relies in the social trust theory mentioned above [56], being therefore in line with the literature.

The Russia's political influence on the Moldova's foreign policy attitudes is transmitted *inter alia* by the Russian communication channels, and this is clearly confirmed by the negative sign of all coefficients which are significant in all four models. People trusting the Russian communication channels strongly reject all political measures which however are rejected by the Russia governments as well [37]. The exception at this point could be the measure of federalization.

The level-2 explanatory variables are less significant in our models compared to the Level-1 variables, but are important for our analysis because they allow also considering the geographical dimension, which would be in line with a large strand of literature on foreign policy attitudes [58]. At the regional Level, the centre-periphery theory frames our results which adds new empirical evidence over the body of papers emphasizing that the "periphery" will "either favour the status-quo, or sudden and complete changes" [27] resulting therefore in a difference in attitudes between the centre and the periphery of the society, as also underlined by [28]. In contrast with the central districts which support the unification with Romania, the South Moldovan districts are found to be in opposition with the EU membership and in favour of federalization.

According to [59], the rural-urban divide, the distance of rural areas from the capital, and the loss of influence in politics in many rural areas have both leaded to different levels of trust and attitudes of population living in rural areas. In our paper, the rural districts have been found agree on the unification with Romania, which is a very sensitive issue on the Chişinău political agenda. The findings at the regional level also reflect the internal political tensions and the spatial dimension of the foreign policy attitudes in Moldova.

## Conclusions and policy recommendations

The aim of the paper was to provide a better understanding of foreign policy attitudes in Moldova, with a focus on their relation with different indicators of poverty, ethnicity and other individual- and regional level characteristics. The multilevel analysis enabled us to provide a regional picture of foreign policy attitudes which fits the specific peculiarities of a country divided by ethnic, regional and political conflicts.

As already explained, the paper contributes to the literature twofold. First, it addresses for the first time the issue of foreign policy attitudes in Moldova, and moreover it frames the analysis in a regional dimension. The existing strand of literature on foreign policy attitudes mainly explores the militant and cooperative internationalism having the US as case-study. Extending the analysis of foreign policy attitudes across other contexts and countries, but on the same axe of cooperative internationalism- militant internationalism- isolationism, facilitates the international comparability of results, interpretation and policies. In this regard, our study follows the line of research developed by [18, 19], and it adds empirical evidence over issues previously formulated by [4]. Secondly, our paper introduces for the first time poverty in the study of public opinions on foreign policy, which allows considering the impact of a prevailing social issue in Moldova.

The most important findings will be resumed here, but an extensive analysis of all empirical results is done in the previous section. Some policy recommendations will be formulated at the end of the section.

Although the unification with Romania and the EU membership represent two distinct foreign policy measures, the impact and significance of their individual level- correlates are almost similar, which proves that becoming member of the EU or a part of Romania are similarly perceived by Moldovans. One explanation is that Romania is an EU member. However, it is interesting to note at this point that Romania is not seen as a big "grabbing" country, but rather as a step toward the EU membership.

In comparison with the other foreign policy measures analysed here, the NATO membership appears as a difficult policy decision for Moldova, given the lack of significance of most of its correlates. Nevertheless, individuals who are satisfied with Moldova's foreign policy are found to also agree with the NATO membership, which might suggest that the supporters of NATO membership perceive this policy measure as an extension of the current foreign policy.

The empirical results reflect a number of common patterns behind all foreign policy measures analysed here. The Russian media influence, being Gagauz, and the opposition to the government decision to eventually help refugees, are found to be all strongly associated with the disapproval of all foreign policy measures. The ethnic identity plays an important role in explaining foreign policy attitudes because, beside Gagauzians who strongly reject all foreign policies, Moldovans agree with the EU membership. The ethnic dimension is even further deepened by the regional dimension, which reflects the geographical perspective. This analysis unveils that rural population supports the idea of unification with Romania, as well as the Centre region which is "more rural" compared to the other regions, while the Southern districts, which concentrate a higher proportion of Gagauzians, disagree with the EU membership and support the idea of federalization.

The relationship between foreign policy attitudes and poverty was at the core of empirical investigation. Different types of poverty indicators have been comparatively analysed, and to facilitate a broader conceptualisation of poverty, a score of deprivation (multidimensional poverty) has been derived from a number of seven deprivation items. Compared to the other poverty measures, deprivation is found to be the only one being significantly related with negative foreign policy attitudes in three models (excepting federalisation). We place this finding in the framework of the "centre-periphery theory" [27], interpreted under the umbrella of peripheralization, which emphasizes the political scepticism of the "social" periphery, represented here by the deprived population.

Although our paper is not primarily aimed to examine whether the foreign policy attitudes in Moldova are structured or not, the overall empirical results, and especially the similarity in the impact that most correlates have on all foreign policy attitudes models, indicate their lack of structure and consistency. This is in line with most findings in the literature on foreign policy attitudes [5, 12, 22].

The usefulness of the paper results go beyond the area of the domestic policy design. In the last decades, the role of public opinion in the structuring of EU foreign policy has increased and has become more complex, as a consequence of the growing role of the European foreign and security policy facing more and more challenges without and within the EU borders. Moldova is a potential new candidate for EU membership. In this light, understanding the foreign policy attitudes in Moldova could provide valuable insights for the EU policy makers, analysts and strategists. Equally, exploring a new dataset on a non-EU country oscillating between Russia and the EU could provide new insights over a small country, insufficiently explored in the international literature.

The links between foreign policy attitudes, ethnicity, regionalism and poverty in Moldova could be further examined in more details, and one theory allowing to capture all these issues into a unitary framework could be the "centre-periphery" theory. This could be a future direction in the analysis of foreign policy attitudes in Moldova that can bring additional empirical insights with useful implications in the economic and social policy area.

## Supporting information

**S1 Dataset. Barometer of public opinion in Moldova 2017.**
(DTA)

## Author Contributions

**Conceptualization:** Monica Răileanu Szeles.

**Data curation:** Monica Răileanu Szeles.

**Formal analysis:** Monica Răileanu Szeles.

**Methodology:** Monica Răileanu Szeles.

**Software:** Monica Răileanu Szeles.

**Visualization:** Monica Răileanu Szeles.

**Writing – original draft:** Monica Răileanu Szeles.

**Writing – review & editing:** Monica Răileanu Szeles.

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
