## [Decision Letter · Decision Letter 0]

2 Nov 2020

PONE-D-20-29273

A regional approach to foreign policy attitudes, poverty and ethnicity in Moldova

PLOS ONE

Dear Dr. Raileanu Szeles,

Thank you for submitting your manuscript to PLOS ONE. After careful consideration, we feel that it has merit but does not fully meet PLOS ONE’s publication criteria as it currently stands. Therefore, we invite you to submit a revised version of the manuscript that addresses the points raised during the review process.

You should revise the manuscript according to all the reviewers' comments. In particular:

Title and abstract should be rewritten to offer a more comprehensive view of the performed research.Justify the chosen variables in accordance with the literature on this issue.Provide details on data availabilityRevise the text, to clarify some misleading sentences as advised by one of the reviewers.

We look forward to receiving your revised manuscript.

Kind regards,

María Carmen Díaz Roldán, Ph.D Economics

Academic Editor

PLOS ONE

Journal Requirements:

2.We note that [Figure(s) 1 and 3] in your submission contain map images which may be copyrighted. All PLOS content is published under the Creative Commons Attribution License (CC BY 4.0), which means that the manuscript, images, and Supporting Information files will be freely available online, and any third party is permitted to access, download, copy, distribute, and use these materials in any way, even commercially, with proper attribution. For these reasons, we cannot publish previously copyrighted maps or satellite images created using proprietary data, such as Google software (Google Maps, Street View, and Earth). For more information, see our copyright guidelines: http://journals.plos.org/plosone/s/licenses-and-copyright.

1.    You may seek permission from the original copyright holder of Figure(s) [1 and 3] to publish the content specifically under the CC BY 4.0 license. 

3. Please ensure that you refer to Figure 2 in your text as, if accepted, production will need this reference to link the reader to the figure.

Reviewers' comments:

Reviewer's Responses to Questions

**Comments to the Author**

1. Is the manuscript technically sound, and do the data support the conclusions?

Reviewer #1: Yes

Reviewer #2: Yes

2. Has the statistical analysis been performed appropriately and rigorously? 

Reviewer #1: Yes

Reviewer #2: I Don't Know

3. Have the authors made all data underlying the findings in their manuscript fully available?

Reviewer #1: Yes

Reviewer #2: No

4. Is the manuscript presented in an intelligible fashion and written in standard English?

Reviewer #1: Yes

Reviewer #2: Yes

5. Review Comments to the Author

Reviewer #1: General Comments

The paper titled “A regional approach to foreign policy attitudes, poverty and ethnicity in Moldova” deals with an interesting topic, focused on a well-centred perspective. It is under the scope of the Journal and, in my view, it provides a better understanding of foreign policy attitudes in Moldova, with a focus on their relation with different indicators of poverty, ethnicity and other individual- and regional level characteristics. It constitutes an interesting focus for reflection.

This work is well written and easily understandable. The structure of the paper, the revision of the literature and the method are well designed. Results and conclusions are very well supported by the literature as well as the empirical analysis.

I give some pieces of advice to the authors to improve the paper, particularly for providing a more detailed explanation in some sections. I think that this more comprehensive explanation will contribute to the fluency of reading as well as to the understanding of the econometric modelling.

Specific Comments

When the two-level variance components model is explained in section 3. Methods, the authors provide a description of all components of equation (1), except the component referred to the group level residual. Although this becomes perfectly clear when you continue reading the work, it is convenient to specify it the first time it is cited.

In the paragraph in which the main variables of interest are described (section 4. Ethno-geopolitical context and Data, in the section dedicated to data), it would be interesting to directly justify the variables chosen according to the academic literature referred to in the previous sections.

I want to point out a minor format matter in the last paragraph of the third section (Methods) since there is a different font format to the rest of the paper.

Reviewer #2: The flow of information throughout the paper is well structured. Very informative piece of work

The main contributions of the submitted article are:

(1) It provides a clear analysis of public opinion on foreign policy for Moldova. It adds an alternative view to the literature in its field since most of the studies are US-based.

(2) It also connects regional patterns and socio-economics issues such as poverty and ethnicities with public perception on various foreign policies currently present in the Moldovan society.

Additional Comments:

- The title of the paper is misleading based on the scope of the article.

- Abstract is hard to read. It does not provide a clear path of what the paper wants to achieve.

- Please, provide more details regarding data availability within the manuscripts and supporting information files. This is not clear at all.

- Kindly, provide a brief context on the Transnistria conflict within your article. Why the conflict was not briefly described in the paper?

- Page 8: Attention to details. References used in the preparation of the paper were not found in the references section. For instance, See Baltag (2013)

- Page 8: It is not clear to readers, what is the pro-poor economic growth of 5%. Please, clarify or add more details.

- Page 8: What do you mean by regional perspective? At this stage, It is not very clear to readers. Please, provide more details.

- Typo on page 8: "the most important ethnic groups in Moldova ae Romanians"

- Page 10: "but rather as latent values explaining the factor loadings according to the factor analysis". Kindly, provide more details, since it is not clear to readers.

- Page 12: " then the use of multilevel models becomes totally justified" Add references to support your approach.

- Page 14: " In the equation below". Shouldn't be in the equation above?

- Page 16. Provide some context/details about the use of Age square in your analysis.

- Typo on page 17: "patters"

- Typo on page 19: "detoted"

- Page 19: Please, provide clarification, details for CTT

- Page 21: is TV missing from Fig 2? If so, there is inconsistency with comments and table 2 on Page 20.

- Page 25: By reading the following "However, the NATO membership is not a seen as an appropriate foreign policy measure for Moldova, not by Moldovans, and nor by Gagauz people, and one explanation could be the strong Russian opposition (Schelegel, 2018).

Aren't Gagauz people Moldovan citizens? The paragraph could imply a different interpretation. Please, clarify.

- Typo on page 25: "are in opposition with the all policy" Please, remove 'the'

- Page 26: The information provided about attitudes toward refugees is almost negligible. Add more text regarding refugees in Moldova.

6. PLOS authors have the option to publish the peer review history of their article (what does this mean?). If published, this will include your full peer review and any attached files.

Reviewer #1: No

Reviewer #2: No

---

## [Author Response · Author response to Decision Letter 0]

1 Dec 2020

Response to reviewers

Reviewer #1: General Comments

1.1 When the two-level variance components model is explained in section 3. Methods, the authors provide a description of all components of equation (1), except the component referred to the group level residual. Although this becomes perfectly clear when you continue reading the work, it is convenient to specify it the first time it is cited.

R. I introduced a short explanation about the group level error (eq.1, section 3)

1.2 In the paragraph in which the main variables of interest are described (section 4. Ethno-geopolitical context and Data, in the section dedicated to data), it would be interesting to directly justify the variables chosen according to the academic literature referred to in the previous sections.

I want to point out a minor format matter in the last paragraph of the third section (Methods) since there is a different font format to the rest of the paper.

R. I introduced a new paragraph (p.10) to justify the selection of variables according to the literature. The selection of the four dependent variables as vectors of foreign policy is extensively motivated in several sections as Introduction and the first sub-section of section 4 (entitled The substance of foreign policy attitudes in Moldova). 

I checked the last paragraph of the third section (Methods) and I didn’t find a different font format to the rest of the paper (TNR12).

Reviewer #2: 

 - The title of the paper is misleading based on the scope of the article.

R. I reformulated the title of the paper.

- Abstract is hard to read. It does not provide a clear path of what the paper wants to achieve.

R. I reformulated some parts of the Abstract.

- Please, provide more details regarding data availability within the manuscripts and supporting information files. This is not clear at all.

R. A large paragraph about the data availability and dataset description is the first one at the beginning of section Data (p.9). The dataset is freely provided online by the Institute for Public Policies in Moldova (as also mentioned in the paper).

- Kindly, provide a brief context on the Transnistria conflict within your article. Why the conflict was not briefly described in the paper?

R. I included a brief presentation of the Transnistria conflict at p.9 of the section “The substance of foreign policy attitudes in Moldova”.

- Page 8: Attention to details. References used in the preparation of the paper were not found in the references section. For instance, See Baltag (2013)

R. Yes, I corrected.

- Page 8: It is not clear to readers, what is the pro-poor economic growth of 5%. Please, clarify or add more details.

R. A footnote was added to provide an additional explanation in this regard.

- Page 8: What do you mean by regional perspective? At this stage, It is not very clear to readers. Please, provide more details.

R. An additional explanation was added in Introduction at p.3.

- Typo on page 8: "the most important ethnic groups in Moldova ae Romanians"

R. I made the correction.

- Page 10: "but rather as latent values explaining the factor loadings according to the factor analysis". Kindly, provide more details, since it is not clear to readers.

R. I reformulated a little bit for more clarity.

- Page 12: " then the use of multilevel models becomes totally justified" Add references to support your approach.

R. I added a reference in this regard.

- Page 14: " In the equation below". Shouldn't be in the equation above?

R. Yes, thank you! I made the correction.

- Page 16. Provide some context/details about the use of Age square in your analysis.

R. I added an explanation in this regard at p.19 (at the point where explaining the effect of age).

- Typo on page 17: "patters"

R. I made the correction.

- Typo on page 19: "detoted"

R. I made the correction.

- Page 19: Please, provide clarification, details for CTT

R. The ICC is described at p.15 (just above Fig.2).

- Page 21: is TV missing from Fig 2? If so, there is inconsistency with comments and table 2 on Page 20.

R. I made the correction.

- Page 25: By reading the following "However, the NATO membership is not a seen as an appropriate foreign policy measure for Moldova, not by Moldovans, and nor by Gagauz people, and one explanation could be the strong Russian opposition (Schelegel, 2018).

Aren't Gagauz people Moldovan citizens? The paragraph could imply a different interpretation. Please, clarify.

R. I introduced a new paragraph providing more information about the Gaguz people, just above the paragraph cited by reviewer.

- Typo on page 25: "are in opposition with the all policy" Please, remove 'the'

R. I made the correction.

- Page 26: The information provided about attitudes toward refugees is almost negligible. Add more text regarding refugees in Moldova.

R. I added more text regarding refugees in Moldova.

---

## [Decision Letter · Decision Letter 1]

29 Dec 2020

Examining the foreign policy attitudes in Moldova

PONE-D-20-29273R1

Dear Dr. Raileanu Szeles,

We’re pleased to inform you that your manuscript has been judged scientifically suitable for publication and will be formally accepted for publication once it meets all outstanding technical requirements.

Kind regards,

María Carmen Díaz Roldán, Ph.D Economics

Academic Editor

PLOS ONE

Additional Editor Comments (optional):

Reviewers' comments:

Reviewer's Responses to Questions

**Comments to the Author**

1. If the authors have adequately addressed your comments raised in a previous round of review and you feel that this manuscript is now acceptable for publication, you may indicate that here to bypass the “Comments to the Author” section, enter your conflict of interest statement in the “Confidential to Editor” section, and submit your "Accept" recommendation.

Reviewer #2: All comments have been addressed

2. Is the manuscript technically sound, and do the data support the conclusions?

Reviewer #2: Yes

3. Has the statistical analysis been performed appropriately and rigorously? 

Reviewer #2: Yes

4. Have the authors made all data underlying the findings in their manuscript fully available?

Reviewer #2: Yes

5. Is the manuscript presented in an intelligible fashion and written in standard English?

Reviewer #2: Yes

6. Review Comments to the Author

Reviewer #2: The flow of information throughout the paper makes this piece of work easy to follow. Very informative and well-structured research manuscript.

Few typos were identified, particularly in the new paragraphs added to the latest version of the manuscript.

7. PLOS authors have the option to publish the peer review history of their article (what does this mean?). If published, this will include your full peer review and any attached files.

Reviewer #2: No

---

## [Editor Report · Acceptance letter]

30 Dec 2020

PONE-D-20-29273R1 

Examining the foreign policy attitudes in Moldova 

Dear Dr. Răileanu Szeles:

I'm pleased to inform you that your manuscript has been deemed suitable for publication in PLOS ONE. Congratulations! Your manuscript is now with our production department. 

Kind regards, 

on behalf of

Dr. María Carmen Díaz Roldán 

Academic Editor

PLOS ONE